# *Fusarium* Keratitis in Taiwan: Molecular Identification, Antifungal Susceptibilities, and Clinical Features

**DOI:** 10.3390/jof8050476

**Published:** 2022-05-03

**Authors:** Tsung-En Huang, Jie-Hao Ou, Ning Hung, Lung-Kun Yeh, David Hui-Kang Ma, Hsin-Yuan Tan, Hung-Chi Chen, Kuo-Hsuan Hung, Yun-Chen Fan, Pei-Lun Sun, Ching-Hsi Hsiao

**Affiliations:** 1Department of Ophthalmology, Chang Gung Memorial Hospital, Linkou Branch, Taoyuan 333, Taiwan; jj791129@gmail.com (T.-E.H.); shsk1212@gmail.com (N.H.); lkyeh@ms9.hinet.net (L.-K.Y.); davidhkma@yahoo.com (D.H.-K.M.); tanhsin@gmail.com (H.-Y.T.); mr3756@cgmh.org.tw (H.-C.C.); agarlic2000@gmail.com (K.-H.H.); 2Department of Plant Pathology, National Chung Hsing University, Taichung 402, Taiwan; allenstorm2005@gmail.com; 3College of Medicine, Chang Gung University, Taoyuan 333, Taiwan; 4Department of Dermatology and Research Laboratory of Medical Mycology, Chang Gung Memorial Hospital, Linkou Branch, Taoyuan 333, Taiwan; fan01290129@gmail.com

**Keywords:** *Fusarium* keratitis, molecular identification, antifungal susceptibility

## Abstract

We performed molecular identification and antifungal susceptibilities of pathogens and investigated clinical features of 43 culture-proven *Fusarium* keratitis cases from 2015–2020 in Taiwan. The pathogens were identified by sequencing of their internal transcribed spacer regions of ribosomal DNA and translation elongation factor 1α gene; their antifungal susceptibilities (to seven agents) were determined by broth microdilution method. We also collected clinical data to compare the drug susceptibilities and clinical features of *Fusarium solani* species complex (FSSC) isolates with those of other *Fusarium* species complexes (non-FSSC). The FSSC accounted for 76.7% pathogens, among which *F. falciforme* (32.6%) and *F. keratoplasticum* (27.9%) were the most common species. Among clinically used antifungal agents, amphotericin B registered the lowest minimal inhibitory concentration (MIC), and the new azoles efinaconazole, lanoconazole and luliconazole, demonstrated even lower MICs against *Fusarium* species. The MICs of natamycin, voriconazole, chlorhexidine, lanoconazole, and luliconazole were higher for the FSSC than the non-FSSC, but no significant differences were noted in clinical outcomes, including corneal perforation and final visual acuity. In Taiwan, the FSSC was the most common complex in *Fusarium* keratitis; its MICs for five tested antifungal agents were higher than those of non-FSSC, but the clinical outcomes did not differ significantly.

## 1. Introduction

*Fusarium* species are the most frequent causative agent of fungal keratitis, a corneal infection that can lead to severe vision loss [1,2]. *Fusarium* keratitis risk factors include contact lens use, ocular trauma, ocular surgery, topical steroid use, and immunosuppression [3]. With increasing numbers of contact lens wearers, *Fusarium* keratitis is becoming a critical health concern worldwide. In 2005, an outbreak of contact lens-associated *Fusarium* keratitis was reported in multiple regions, including Hong Kong, Singapore, Europe, and the United States (US) [4,5,6]. This event was reportedly associated with decreased antimicrobial activity in Bausch and Lomb’s contact lens cleaning solution ReNu with MoistureLoc [7]. *Fusarium* keratitis is an important issue in multiple countries, and Taiwan is no exception. *Fusarium* species account for 44.6% of the filamentous fungal keratitis in Taiwan [3], and the proportion of *Fusarium*-related cases among all microbial keratitis increased from 4.0% to 6.4% between 1992 and 2016 [8].

Early diagnosis and prompt delivery of antifungal agents are key factors in *Fusarium* keratitis treatment. Although traditional morphology is frequently adopted in daily clinical practice for diagnosis, it is time-consuming and cannot be used to differentiate isolates to the species level. In contrast, molecular diagnosis has advantages of expediency, accuracy, and genotyping capacity. Molecular identification is crucial to epidemiology and can influence therapy and outcomes in cases of interspecies differences in antifungal susceptibilities or virulence. However, despite those advantages, molecular diagnosis was still not widely used around the world. From a recent meta-review, only 11.9% of the *Fusarium* isolates (*n* = 628) were identified to species level among patients with *Fusarium* keratitis (*n* = 5294) [9].

Epidemiological studies involving molecular diagnosis of *Fusarium* keratitis have been conducted in several countries, including Germany, the Netherlands, India, and Brazil [10,11,12,13]. However, no such studies have been performed in East Asia. Because the epidemiological patterns may vary by country and area, we performed molecular identification and antifungal susceptibilities of *Fusarium* species in Taiwan. In addition, studies of the correlations of *Fusarium* species and antifungal susceptibilities with clinical outcomes are rare; therefore, we investigated such correlations by comparing members of the *F. solani* species complex (FSSC) with those of other *Fusarium* (i.e., non-FSSC) species.

## 2. Materials and Method

### 2.1. Study Population and Data Collection

We extracted information regarding culture-proven *Fusarium* keratitis cases from 1 January 2015, through 31 December 2020, from the database of the microbiological laboratory at CGMH, Linkou branch, Taiwan. We obtained cornea scrapings from patients with infectious keratitis and sent specimens to smear and culture examination. The cornea scraping specimens were cultivated on blood and chocolate agar, modified Sabouraud agar, Lowenstein–Jensen agar slants, and thioglycolate broth. Positive *Fusarium* culture was defined as *Fusarium* species growth identified through morphology on two media, fungal elements observed in smears and *Fusarium* species growth on one medium, or confluent growth of *Fusarium* species on one medium.

We reviewed the medical charts of the patients associated with the keratitis cases for demographic data, systemic and ocular disease history, predisposing factors, initial ocular presentation, antifungal treatment, surgery requirement, hospitalization, corneal perforation or endophthalmitis, and initial and final visual acuity (VA). The predisposing factors were ocular trauma, an outdoor occupation or gardening habit, contact lens use, topical steroid use, preexisting ocular disease, and recent ocular surgery. Preexisting ocular disease referred to any disease that interfered with the epithelial integrity of the cornea. Recent ocular surgery meant any surgery performed up to 3 months before the ulcer. An ulcer was defined as central if it was less than 2 mm from the corneal center and peripheral if it extended to within 2 mm of the limbus; otherwise, the ulcer was defined as paracentral if its location was between the definition of cornea center and periphery. If the epithelium defect covered the cornea from center to periphery in all four quadrants, it was defined as near total epithelium defect. Ulcers less than 2 mm in diameter were considered small, those 2–6 mm in diameter were considered medium, and those larger than 6 mm were considered large. As a clinical outcome, corneal perforation also included cases of impending perforation that required therapeutic penetrating keratoplasty (TPK) or amniotic membrane transplantation (AMT). VA was measured using Snellen charts and converted to the logarithm of minimum angle of resolution (logMAR) VA. As suggested by Schulze-Bonsel et al. [14], nonnumerical VA data were converted to logMAR values as follows: counting fingers = 2, hand motion = 2.3, light perception = 2.7, and no light perception = 3.0. Initial VA was measured on the first visit for *Fusarium* keratitis. Final VA, if available, was measured as best corrected VA (BCVA) on the final visit.

### 2.2. DNA Extraction, Amplification, and Sequencing

All the *Fusarium* isolates underwent molecular diagnosis process. If one patient received multiple times of corneal fungal culture during the treatment period, the *Fusarium* isolates each time would all undergo molecular diagnosis individually and be given their own strain number.

Fungal isolates were subcultured on potato dextrose agar for purification. Thereafter, part of the mycelium was collected and placed in a plastic vial, to which 0.13 g of metal beads and 800 μL of lysis buffer (Tris buffer, surfactants, pH 8.0) were added. The vial was then transferred to a cell disruptor (Mini-BeadBeater 16, BioSpec, Bartlesville, OK, USA) to break down the cell walls. The buffer fluid containing the fungal fragments was transferred to a DNA extraction kit, and genomic DNA was extracted with a Smart LabAssist (TANBead, Taiwan) automatic DNA extraction system. Internal transcribed spacers of ribosomal DNA (ITS) and translation elongation factor-1*α* gene (*TEF-1α*) were used for molecular identification. The ITS regions were amplified with primers ITS1 (TCCGTAGGTGAACCTGCGG) and ITS4 (TCCTCCGCTTATTGATATGC), and *TEF-1α* gene was amplified with primers EF1 (ATGGGTAAGGARGACAAGAC) and EF2 (GGARGTACCAGTSATCATG). The PCR conditions were as described previously [15,16]. The PCR products were confirmed by gel electrophoresis, then purified and submitted for DNA sequencing with ABI Prism 3730 xl DNA analyzer (Applied Biosystems, Foster City, CA, USA). The sequences were uploaded to *Fusarium* MLST database at Mycobank website (https://fusarium.mycobank.org/, accessed on 1 February 2022) and *Fusarium* Database (http://isolate.fusariumdb.org, accessed on 1 February 2022) for preliminary species identification. All sequences generated in this study were uploaded to the DNA Data Bank of Japan (DDBJ) [17].

### 2.3. Phylogenetic Analyses

Based on the preliminary identification results from the *Fusarium* MLST database, the ITS and *TEF-1α* sequences of allied *Fusarium* species were retrieved from GenBank (Table 1). Each region was first aligned with MAFFT online version [18], and poorly aligned regions were removed manually or with Gblocks online software [19]. DNA evolution models were determined with Smart Model Selection (SMS) [20]. The two loci were then concatenated for the following phylogenetic analyses. Maximum likelihood trees were inferred by RAxML-NG v. 1.1.0, and Felsenstein’s bootstrap (FBP) statistical support was calculated from 1000 resample data sets [21]. Bayesian inference tree was inferred by using MrBayes v3.2.6 x64 [22]. The analysis was performed with two MCMC chains for 1,000,000 generations, and one tree was sampled for every 1000 generations. After discarding the first 250 trees (burn-in), a consensus tree was obtained from the remaining 750 trees. All analyses were conducted on the Linux Mint 20.3 (64-bit) operating system, and to ensure reproducibility, the random seeds were explicitly set to 56 wherever they were needed.

### 2.4. Antifungal Susceptibility Testing

Broth microdilution was used to determine the minimal inhibitory concentration (MIC) of antifungals to various strains. The procedures were performed in accordance with the third edition of *M38: Reference Method for Broth Dilution Antifungal Susceptibility Testing of Filamentous Fungi*, published by the Clinical and Laboratory Standards Institute [23]. Pure powders of antifungals and chemical for testing were purchased from Sigma-Aldrich^®^ and the range of concentration was amphotericin B (16–0.031 μg/mL), natamycin (32–0.063 μg/mL), voriconazole (16–0.031 μg/mL), chlorhexidine (256–0.5 μg/mL), efinaconazole (4–0.008 μg/mL), lanoconazole (0.5–0.001 μg/mL), and luliconazole (0.5–0.001 μg/mL). ATCC 22,019 *Candida parapsilosis* and ATCC 6258 *Candida krusei* were used as control. MIC endpoints were determined using a reading mirror after 48 h of incubation at 35 °C and indicated by a 100% inhibition of growth compared with drug-free growth control well for all the drugs and chemicals.

If one patient received corneal fungal culture multiple times and two or more isolates were obtained, each *Fusarium* isolate was subjected to antifungal susceptibility tests individually. However, we only used the drug susceptibility results from the first *Fusarium* isolates when presenting the antifungal susceptibilities or comparing the MIC between different *Fusarium* species complex.

### 2.5. Statistical Analysis

The isolates were separated by molecular identification into the FSSC and non-FSSC groups. The antifungal susceptibilities and clinical features of these two groups were compared. The initial and final logMAR VA values were compared using Mann–Whitney *U* tests. Categorical demographic and outcome variables were compared using chi-square tests; continuous variables in demographic data, clinical outcomes, and MIC results were compared using Student’s *t* test. Statistical significance was defined as *p* < 0.05. All statistical analyses were performed using SPSS version 25 (IBM, New York, NY, USA). 

## 3. Results

### 3.1. Molecular Identification of Fusarium Isolates

A total of 43 cases of culture-proven *Fusarium* keratitis from the study period of 2015–2020 were analyzed. Four out of the 43 patients received multiple corneal scrapings for fungal culture during the treatment period, so eventually, 52 *Fusarium* isolates were obtained and underwent molecular diagnosis. Based on multilocus phylogenetic analyses, 14 *Fusarium* species, belonging to five species complexes, were identified and the result is shown in Table 1 and Figure 1. For those four patients receiving multiple fungal culture, their pathogens remained the same throughout the sampling period.

Three isolates (CGMHD 1660, 2436, and 3049) were clustered with NRRL 37,393 and CBS 110,307 in *F. dimerum* species complex (FDSC), and these strains had already been identified by Hans-Josef Schroers et al., in 2009, but it still remained unnamed. Hence, we presented these three isolates as *Fusarium* sp. FDSC in this article [24]. The species of CGMHD 2044 isolate was still uncertain, because its *TEF-**1**α* result had only 96.3% similarity to an isolate of *F. incarnatum* in Genbank (MAFF 236386), but in another article, the same speceis was renamed as *F. semitectum* [25]. Hence, further species identification was required for this isolate and it was only presented as *F**usarium* sp. FIESC in this article.

The pathogen distribution in 43 patients with keratitis based on molecular identification is shown in Table 2. Thirty-three cases (76.7%) belonged to the FSSC group, among which 14 (32.6%) were *F. falciforme*, and 12 (27.9%) were *F. keratoplasticum*. Four cases (9.3%) belonged to the FDSC, among which three (7.0%) were *F. dimerum* SC and one (2.3%) was *F. delphinoides*. The *F. oxysporum* species complex (FOSC), *F. incarnatum-equiseti* species complex (FIESC), and *F. fujikuroi* species complex (FFSC) each accounted for two cases (4.7%). For data analysis, those cases were divided into the FSSC and non-FSSC groups.

### 3.2. Antifungal Susceptibilities

As indicated in Table 3, we compared the geometric mean (GM) MICs of each antifungal agent against the FSSC and non-FSSC isolates. Among the antifungal agents currently used in clinical practice (i.e., amphotericin B, natamycin, voriconazole, and chlorhexidine), amphotericin B had the lowest GM MIC against both the FSSC and non-FSSC isolates, but the new azoles, including efinaconazole, lanoconazole, and luliconazole, demonstrated even lower GM MICs for *Fusarium* species. The GM MICs of natamycin (*p* = 0.021), voriconazole (*p* = 0.010), chlorhexidine (*p* < 0.001), lanoconazole (*p* = 0.004), and luliconazole (*p* = 0.026) were higher for the FSSC isolates than the non-FSSC isolates. In addition, the FSSC group also exhibited a trend of higher MIC in efinaconazole (*p* = 0.052). The detailed MIC data for each isolate are presented in the Appendix A, Table A1.

### 3.3. Demographic Data, Predisposing Factors, and Initial Presentation

As listed in Table 4, the study sample comprised cases from 29 (67.4%) men and 14 (32.6%) women. Their average age was 51.5 ± 19.5 (11–91) years. No differences between the FSSC and non-FSSC groups were observed in terms of age or sex. Outdoor occupation or gardening habit (*n* = 23, 53.5%) and ocular trauma (*n* = 19, 44.2%) were the most common predisposing factors, followed by recent ocular surgery (*n* = 5, 11.6%), contact lens use (*n* = 4, 9.3%), preexisting ocular disease (*n* = 4, 9.3%), and topical steroid use (*n* = 1, 2.3%). The proportion of contact lens use was significantly higher in the non-FSSC group than in the FSSC group (30% vs. 3%, *p* = 0.010), but no significant differences were noted in the other predisposing factors. Most of the corneal ulcers were in the paracentral area (*n* = 27, 62.8%) and of medium size (*n* = 27, 62.8%). No significant differences in ulcer area or location were observed between the FSSC and non-FSSC groups. At initial presentation, 23 of the case patients (53.5%) had hypopyon, but none exhibited corneal perforation.

### 3.4. Treatment and Outcomes

As listed in Table 5, among the 43 patients, 29 (66.7%) required hospitalization, for 12.3 ± 15.3 days on average. All the patients were treated with antifungal agents, for an average 31.8 ± 33.4 days. The most used topical antifungal agents were natamycin (44.2%), voriconazole (44.2%), and amphotericin B (39.5%), which were used either as single agents or in combination. Twenty-three patients (53.5%) underwent surgical procedures, including keratectomy (*n* = 12, 27.9%), AMT (*n* = 4, 9.3%), and TPK (*n* = 7, 16.3%). Two patients developed endophthalmitis (4.7%), and ten experienced corneal perforation (23.3%). Final VA was worse than 20/200 in 15 patients (34.9%). No significant differences were observed between the FSSC and non-FSSC groups in treatment, complications, or visual outcomes.

## 4. Discussion

We investigated the epidemiological data of *Fusarium* isolates from 43 cases of *Fusarium* keratitis diagnosed at a referral center in Taiwan. We identified 13 *Fusarium* species belonging to five species complexes; 33 of the isolates (76.7%) belonged to the FSSC. We also tested susceptibilities to seven antifungals, among which amphotericin B achieved the lowest MICs. The GM MICs of natamycin, voriconazole, chlorhexidine, lanoconazole, and luliconazole were significantly higher for FSSC isolates than for non-FSSC isolates. An outdoor occupation or gardening habit and ocular trauma were the most common predisposing factors; 23 patients (53.5%) required surgical procedures, including ten (23.3%) operations for perforation, and 15 patients (34.9%) had final VA of less than 20/200. No significant differences in clinical features were observed between the FSSC and non-FSSC groups.

The FSSC accounted for more than three-quarters of the *Fusarium* keratitis cases, a finding consistent with those of epidemiological studies of *Fusarium* keratitis in other countries [10,11,12,13,26,27,28]. Although the FSSC was the leading cause of *Fusarium* keratitis, the proportion of FSSC-related keratitis was variable in different countries. It has been reported as high as 75–88% in tropical or subtropical areas such as India, Brazil, or Miami (US) [12,13,26]. By contrast, the proportion of FSSC-related keratitis cases reported in temperate countries such as Germany and the Netherlands has been 41–59% [10,11].

In our study, the most common predisposing factors were an outdoor occupation or gardening habit and ocular trauma; these accounted for approximately half of the cases. By contrast, contact lens use was a factor in only 9.3% of cases. Because our hospital is a regional referral center near industrial and agricultural areas, our results are similar to those reported in countries with large agricultural or industrial sectors, such as Mexico and Brazil, in which ocular trauma is the most common risk factor (57.4% and 48.8%, respectively) and contact lens is implicated in only 8.1% and 4.9% of cases, respectively [12,13]. By contrast, in high-income countries, such as the US, Germany, and the Netherlands, contact lens use is the primary risk factor and associated with 63.5–81.8% of cases, whereas ocular trauma is implicated in only 5.6–17% of cases [10,11,29].

So far, there is still no established clinical breakpoint for antifungal susceptibility test for *Fusarium* species according to European Committee on Antimicrobial Susceptibility Testing (EUCAST) [30]. In our study, amphotericin B registered the lowest GM MIC (FSSC = 1.66 mg/L, non-FSSC = 1.36 mg/L), followed by natamycin (FSSC = 5.79 mg/L, non-FSSC = 3.70 mg/L) and voriconazole (FSSC = 7.29 mg/L, non-FSSC = 2.94 mg/L) among the antifungal agents used to treat *Fusarium* keratitis. Many epidemiological studies have reported comparable results, with amphotericin B exhibiting the lowest mean MIC (1.0–2.0 mg/L); the mean MIC results for natamycin (4.0–6.5 mg/L) and voriconazole (5.0–9.9 mg/L) are also similar to our results [10,11,13,26,27,29]. In the US, Oechesler et al., compared the antifungal susceptibilities of FSSC and non-FSSC species and reported that the MIC of voriconazole was higher for FSSC than for non-FSSC species but observed no differences in the MICs for amphotericin B or natamycin [29], which contradicted with our findings that the MICs of natamycin and voriconazole were higher for the FSSC group but no differences between the MICs of amphotericin B against FSSC and non-FSSC species. The discrepancies in these results are probably due to regional differences or differences in the proportions of *Fusarium* species complexes (with FSSC and FOSC accounting for 75% and 16%, respectively, in the study of Oechesler et al.).

*Fusarium* keratitis–related information on molecular identification, antifungal susceptibilities, and correlations with clinical outcomes is scarce [13,26,27,29], but such information may be helpful for guiding treatment selection and predicting patient outcomes. According to an study in Miami (US), Oechesler et al. reported that FSSC cases were associated with longer recovery times, poorer follow-up BCVA, and a greater need for urgent surgical procedures than were non-FSSC cases; in addition, they suggested that other factors such as interspecies differences in pathogenicity, rather than antifungal susceptibilities, may explain the difference in outcomes between FSSC and non-FSSC case because no difference was observed between groups in the MICs of natamycin or amphotericin B—the most commonly used topical antifungal agents [26]. However, in another study in Brazil by Oechesler et al., they did not obtain the same findings except for the greater need for TPK in FSSC cases than in non-FSSC cases, which may have been associated with a higher natamycin MIC [13]. In our study, despite higher MICs of multiple antifungal agents for the FSSC group, such differences in clinical outcomes were not observed between groups. Again, regional differences or differences in the proportions of *Fusarium* species complexes may explain these discrepant results. In addition, our sample may have been too small for differences to become evident.

Although amphotericin B has had the lowest MIC against *Fusarium* species in multiple studies, no consensus has been reached regarding which antifungal agent is optimal for the treatment of *Fusarium* keratitis. Studies have suggested amphotericin B as the first choice treatment for yeast infection and aspergillosis and natamycin for filamentous fungal keratitis [1,31]. In addition, the absolute MIC can be converted in a relative MIC on the basis of the typically prescribed concentration, as suggested by Lalitha et al., revealing that amphotericin B has a higher relative MIC and may be less likely to be the optimal treatment for *Fusarium* keratitis [32]. For example, the typically prescribed concentration of amphotericin B is 1.5 mg/mL, which is 903 times the MIC for FSSC cases (1.66 mg/L). However, the concentration of natamycin in commercial eyedrops is 5%, which is 3012-fold the MIC for FSSC cases. Based on the typically prescribed concentration, the relative MIC of natamycin is markedly higher than that of amphotericin B. Although we cannot draw conclusions directly from relative MICs because drug bioavailability and corneal penetration must also be considered, the concept of relative MIC provides a new perspective on how antifungal agents operate in clinical practice. In a study of antifungal susceptibilities in fungal keratitis cases in Shandong Province, China, 94.2% of *F. solani* and 91.3% of *F. oxysporum* cases were susceptible to natamycin, whereas only 82.4% of *F. solani* and 74% of *F. oxysporum* cases were susceptible to amphotericin B [33]. Thus, the authors of that study suggested natamycin as the first choice for the treatment of *Fusarium* keratitis. Moreover, amphotericin B’s ocular toxicity and strong irritation of the ocular surface limit its clinical application. In our clinical experience, topical natamycin 5% is the optimal choice for treating *Fusarium* keratitis, but in cases where the clinical response is poor, we recommend adding amphotericin B or voriconazole. However, due to frequent natamycin shortages, we occasionally must begin treatment with amphotericin B or voriconazole instead, explaining why only 44.2% of cases were treated with natamycin (Table 5). The optimal first-line antifungal agent for filamentous fungal keratitis remains debated. As mentioned earlier, although amphotericin B is typically the first choice for conditions caused by Aspergillosis and yeast infections [1,31], its high relative MIC, ocular toxicity, and ocular surface irritation indicate that it may not be the optimal choice. Although voriconazole demonstrates broad-spectrum antifungal activity and favorable corneal penetration, the Mycotic Ulcer Treatment Trial revealed that patients with natamycin-treated fungal keratitis had better 3-month BCVA and a lower risk of corneal perforation than did those with voriconazole-treated cases. However, in *Aspergillus* keratitis, natamycin-treated cases are associated with greater risks of corneal perforation and TPK requirement than voriconazole-treated cases are [34]. Therefore, identifying a new first-line antifungal agent for filament fungal keratitis is essential. Chlorhexidine 0.2% is commonly used in *Acanthamoeba* keratitis treatment, but seldom used as an antifungal agent in clinical practice. A recent evidence-based study recommended that chlorhexidine 0.2% could be an alternative to natamycin in the treatment of filamentous fungal keratitis [35], when natamycin is not available. In Taiwan, we often encountered with shortage of natamycin, so we decided to investigate the chlorhexidine susceptibility in the *Fusarium* isolates. The GM MIC of chlorhexidine in our study was similar to that in the Dutch study by Oliveira dos Santos et al. [36]; they further demonstrated chlorhexidine having fungicidal activity against 90% of *F. oxysporum* strains and 100% of the *F. solani* strains, which supported the clinical efficacy of chlorhexidine. Moreover, we assessed susceptibilities to three new azoles, namely efinaconazole, lanoconazole, and luliconazole; all three drugs demonstrated extremely low MICs against all *Fusarium* species In a previous study, efinaconazole, lanoconazole, and luliconazole also registered low MICs against *Fusarium* isolates from corneal scrapings in vitro, with MIC50 values of 1, 0.06, and 0.03 mg/L, respectively [37]. In addition, these three drugs showed potential effects against a broad spectrum of pathogens in filamentous fungal keratitis, including *Aspergillus* [38], *Purpureocillium* [37], *Beauveria* [37], and *Scedosporium* [39] species. Moreover, because of their low molecular weight and lipophilicity, efinaconazole (348.39 Da), lanoconazole (319.8 Da), and luliconazole (354 Da) have theoretically good penetration into the corneal stroma and anterior chamber. However, currently, none of these new azoles are available as eyedrops. Only luliconazole is clinically available, in the forms of 1% ointment and 1% solution, for treating onychomycosis and dermatophytosis [40]. With low MICs against *Fusarium* species and theoretically good tissue penetration, these new azoles may be favorable candidates for filamentous fungal keratitis treatment in the future.

This study has several limitations. First, the clinical data might be incomplete because of the retrospective study design, and the physicians used diverse protocols in treating patients. Second, in vitro antifungal susceptibility may not be an accurate reflection of the clinical responses to keratitis because the concentration of antifungal agents may remain above the MIC on the ocular surface in cases of high-frequency topical use.

Third, although we incorporated *TEF-1α* as suggested by previous studies for the identification of these clinical *Fusarium* species [41,42], we found that *TEF-1α* alone failed to resolve some closely related species into well-supported monophyletic clades (e.g., *F. brevis* and *F. vanettenii*). In addition, some strains could not be classified as any known species in the ITS + *TEF-1α* analysis (e.g., CGMHD2044). Further investigation is required to determine whether these isolates are new *Fusarium* species. Moreover, our sample size was small, especially the non-FSSC group. We plan to include more cases to compare the clinical outcomes of FSSC and non-FSSC cases with greater statistical power in the future. Finally, as with other microbiological studies, our results may not be generalizable to other populations or territories because our study was conducted only at a single referral center in Taiwan. Nevertheless, this study is the first to provide an overview of *Fusarium* keratitis in Taiwan, including molecular identification, antifungal susceptibilities, and clinical features. Moreover, only limited literature investigated the drug susceptibilities of efinaconazole, lanoconazole, and luliconazole against *Fusarium* keratitis isolates in vitro; our study not only provided supporting results for the potential of these new azoles in *Fusarium* keratitis treatment but also is the first to manifest the difference in their MICs between the FSSC and the non-FSSC isolates in vitro.

## 5. Conclusions

This epidemiological study analyzed *Fusarium* keratitis identified through molecular diagnostics at a referral center in Taiwan. The FSSC was the most common species complex, accounting for 76.7% of the *Fusarium* keratitis cases. In comparison with non-FSSC isolates, the MICs for FSSC isolates were higher for multiple antifungal agents, namely natamycin, voriconazole, chlorhexidine, luliconazole, and lanoconazole, but no significant differences in clinical outcomes were observed between FSSC and non-FSSC cases. Among the antifungal agents used in clinical practice, amphotericin B exhibited the lowest MIC, followed by natamycin and voriconazole. However, the three new azoles, efinaconazole, luliconazole, and lanoconazole, registered even lower MICs against *Fusarium* in vitro. Although these azoles are currently unavailable in eyedrop form, their broad-spectrum antifungal abilities and potentially good tissue penetration make them promising candidates for fungal keratitis treatment in the future.

## Figures and Tables

**Figure 1 jof-08-00476-f001:**
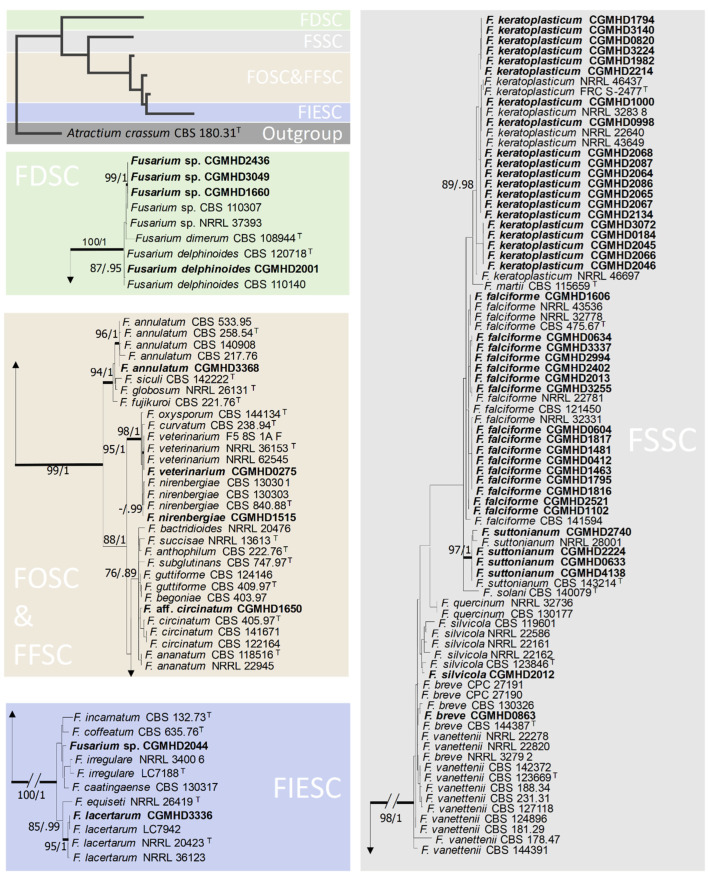
The phylogenetic tree of the 52 *Fusarium* isolates from patients with keratitis. All the *Fusarium* isolates were labelled in bold and strain number were provided.

**Table 1 jof-08-00476-t001:** *Fusarium* species and GenBank accession numbers of *Fusarium* isolates (*n* = 52 *) from patients with keratitis.

*Fusarium* Isolates Strain Number	Species	Accession Number	
		ITS	*TEF-1α*
CGMHD0184	*Fusarium keratoplasticum*	LC683270	LC683322
CGMHD0412	*Fusarium falciforme*	LC683271	LC683323
CGMHD0604	*Fusarium falciforme*	LC683272	LC683324
CGMHD0633	*Fusarium suttonianum*	LC683273	LC683325
CGMHD0634	*Fusarium falciforme*	LC683274	LC683326
CGMHD0820	*Fusarium keratoplasticum*	LC683275	LC683327
CGMHD0863	*Fusarium breve*	LC683276	LC683328
CGMHD0998	*Fusarium keratoplasticum*	LC683277	LC683329
CGMHD1000	*Fusarium keratoplasticum*	LC683278	LC683330
CGMHD1102	*Fusarium falciforme*	LC683279	LC683331
CGMHD1463	*Fusarium falciforme*	LC683280	LC683332
CGMHD1481	*Fusarium falciforme*	LC683281	LC683333
CGMHD1606	*Fusarium falciforme*	LC683282	LC683334
CGMHD1794	*Fusarium keratoplasticum*	LC683283	LC683335
CGMHD1795	*Fusarium falciforme*	LC683284	LC683336
CGMHD1816	*Fusarium falciforme*	LC683285	LC683337
CGMHD1817	*Fusarium falciforme*	LC683286	LC683338
CGMHD1982	*Fusarium keratoplasticum*	LC683287	LC683339
CGMHD2012	*Fusarium silvicola*	LC683288	LC683340
CGMHD2013	*Fusarium falciforme*	LC683289	LC683341
CGMHD2045	*Fusarium keratoplasticum*	LC683290	LC683342
CGMHD2046	*Fusarium keratoplasticum*	LC683291	LC683343
CGMHD2064	*Fusarium keratoplasticum*	LC683292	LC683344
CGMHD2065	*Fusarium keratoplasticum*	LC683293	LC683345
CGMHD2066	*Fusarium keratoplasticum*	LC683294	LC683346
CGMHD2067	*Fusarium keratoplasticum*	LC683295	LC683347
CGMHD2068	*Fusarium keratoplasticum*	LC683296	LC683348
CGMHD2086	*Fusarium keratoplasticum*	LC683297	LC683349
CGMHD2087	*Fusarium keratoplasticum*	LC683298	LC683350
CGMHD2134	*Fusarium keratoplasticum*	LC683299	LC683351
CGMHD2214	*Fusarium keratoplasticum*	LC683300	LC683352
CGMHD2224	*Fusarium suttonianum*	LC683301	LC683353
CGMHD2402	*Fusarium falciforme*	LC683302	LC683354
CGMHD2521	*Fusarium falciforme*	LC683303	LC683355
CGMHD2740	*Fusarium suttonianum*	LC683304	LC683356
CGMHD2994	*Fusarium falciforme*	LC683305	LC683357
CGMHD3072	*Fusarium keratoplasticum*	LC683306	LC683358
CGMHD3140	*Fusarium keratoplasticum*	LC683307	LC683359
CGMHD3224	*Fusarium keratoplasticum*	LC683308	LC683360
CGMHD3255	*Fusarium falciforme*	LC683309	LC683361
CGMHD3337	*Fusarium falciforme*	LC683310	LC683362
CGMHD4138	*Fusarium suttonianum*	LC683311	LC683363
CGMHD0275	*Fusarium veterinarium*	LC683312	LC683364
CGMHD1515	*Fusarium nirenbergiae*	LC683313	LC683365
CGMHD3336	*Fusarium lacertarum*	LC683314	LC683366
CGMHD2044	*Fusarium* sp. FIESC ^a^	LC683315	LC683367
CGMHD1650	*Fusarium aff. circinatum*	LC683316	LC683368
CGMHD3368	*Fusarium annulatum*	LC683317	LC683369
CGMHD1660	*Fusarium* sp. FDSC ^b^	LC683318	LC683370
CGMHD2001	*Fusarium delphinoides*	LC683319	LC683371
CGMHD2436	*Fusarium* sp. FDSC ^b^	LC683320	LC683372
CGMHD3049	*Fusarium* sp. FDSC ^b^	LC683321	LC683373

* Four out of the 43 patients received multiple corneal fungal culture during the treatment period, so eventually, 52 *Fusarium* isolates were collected. All the 52 isolates underwent molecular diagnosis and antifungal susceptibility tests individually. ITS: internal transcribed spacer region; *TEF-1α*: translation elongation factor -1α. ^a^ The *TEF-1α* similarity of this isolate was only 96.3% to an isolate of *F. incarnatum* in Genbank (MAFF 236386), but in another article, the same species was renamed as *F. semitectum* [25]. Hence, further species identification was required for this isolate. ^b^ Three isolates (CGMHD 1660, 2436, and 3049) were clustered with NRRL 37,393 and CBS 110,307 in *F. dimerum* species complex (FDSC), and these strains had already been identified by Hans-Josef Schroers et al., in 2009 but it still remained unnamed [24].

**Table 2 jof-08-00476-t002:** *Fusarium* species distribution in patients with keratitis.

Species Complex Number (%)	*Fusarium* Species	Case Number (%)
FSSC 33 (76.7%)	*F. falciforme*	14 (32.6%)
	*F. keratoplasticum*	12 (27.9%)
	*F. solani.*	3 (7.0%)
	*F. suttonianum*	2 (4.7%)
	*F. silvicola*	1 (2.3%)
	*F. breve*	1 (2.3%)
FDSC 4 (9.3%)	*Fusarium* sp. FDSC ^a^	3 (7.0%)
	*F. delphinoides*	1 (2.3%)
FOSC 2 (4.7%)	*F. nirenbergiae*	1 (2.3%)
	*F. veterinarium*	1 (2.3%)
FIESC 2 (4.7%)	*F. lacertarum*	1 (2.3%)
	*Fusarium* sp. FIESC ^b^	1 (2.3%)
FFSC 2 (4.7%)	*F. aff. circinatum*	1 (2.3%)
	*F. annulatum*	1 (2.3%)

FSSC: *Fusarium solani* species complex; FDSC: *Fusarium dimerum* species complex; FOSC: *Fusarium oxysporum* species complex; FIESC: *Fusarium incarnatum-equiseti* species complex; FFSC: *Fusarium fujikuroi* species complex. ^a^ Hans-Josef Schroers et al., had identified this *Fusarium* species in 2009 but it was still unnamed [24]. ^b^ Further species identification was required for this isolate because its *TEF-1α* similarity was only 96.3% to an isolate of *F. incarnatum* in Genbank (MAFF 236386), but in another article, the same species was renamed as *F. semitectum* [25].

**Table 3 jof-08-00476-t003:** Comparison of antifungal susceptibilities between FSSC (*n* = 33) and non-FSSC (*n* = 10).

Antifungal Agents	*Fusarium* Species	Range of MIC (mg/L)	GM MIC (mg/L)	*p* Value
Amphotericin B	FSSC	0.5~4	1.66	0.263
	Non-FSSC	1~2	1.36	
Natamycin	FSSC	4~16	5.79	0.021
	Non-FSSC	2~8	3.70	
Voriconazole	FSSC	2~16	7.29	0.010
	Non-FSSC	2~8	2.94	
Chlorhexidine	FSSC	16~32	18.38	<0.001
	Non-FSSC	8~16	9.33	
Efinaconazole	FSSC	0.5~2	1.15	0.052
	Non-FSSC	0.25~4	0.46	
Lanoconazole	FSSC	0.125~0.5	0.20	0.004
	Non-FSSC	0.063~0.25	0.09	
Luliconazole	FSSC	0.031~0.25	0.06	0.026
	Non-FSSC	0.016~0.063	0.03	

FSSC: *Fusarium solani* species complex; GM: geometric mean; MIC: minimal inhibitory concentration; non-FSSC: *non- Fusarium solani* species complex.

**Table 4 jof-08-00476-t004:** Demographic data, predisposing factors, and initial presentation of 43 patients with *Fusarium* keratitis.

		Total (%)		FSSC (%)		Non-FSSC (%)		*p*-Value
Gender	Total case number	43		33		10		
	Male	29	(67.4%)	23	(69.7%)	6	(60.0%)	0.566
	Female	14	(32.6%)	10	(30.3%)	4	(40.0%)	
Age	Average (range)	51.5 ± 19.5	(11~91)	50.5 ± 20.0	(11~91)	54.6 ± 18.4	(19~86)	0.572
Predisposing Factors ^a^	Outdoor occupation or gardening habit	23	(53.5%)	18	(54.5%)	5	(50.0%)	0.801
	Trauma	19	(44.2%)	17	(51.5%)	2	(20.0%)	0.079
	Recent ocular surgery	5	(11.6%)	4	(12.1%)	1	(10.0%)	0.855
	Contact lens use	4	(9.3%)	1	(3.0%)	3	(30.0%)	0.010
	Preexisting ocular disease	4	(9.3%)	3	(9.1%)	1	(10.0%)	0.931
	Topical steroid use	1	(2.3%)	1	(3.0%)	0	(0.0%)	0.578
Ulcer location	Central	5	(11.6%)	4	(12.1%)	1	(10.0%)	0.855
	Paracentral	27	(62.8%)	21	(63.6%)	6	(60.0%)	0.835
	Peripheral	7	(16.3%)	4	(12.1%)	3	(30.0%)	0.180
	Near total	4	(9.3%)	4	(12.1%)	0	(0.0%)	0.248
Ulcer area	Small	7	(16.3%)	5	(15.2%)	2	(20.0%)	0.716
	Medium	27	(62.8%)	20	(60.6%)	7	(70.0%)	0.59
	Large	9	(20.9%)	8	(24.2%)	1	(10.0%)	0.332
Hypopyon		23	(53.5%)	19	(57.6%)	4	(40.0%)	0.878

^a^ Total percentage is more than 100% because 18 patients had multiple risk factors. FSSC: *Fusarium solani* species complex; non-FSSC: *non- Fusarium solani* species complex.

**Table 5 jof-08-00476-t005:** Treatment and outcomes of 43 patients with *Fusarium* keratitis.

		Total (%)		FSSC (%)		Non-FSSC (%)		*p*-Value
Hospitalization		29	(66.7%)	22	(66.7%)	7	(67.4%)	
	average days	12.3 ± 15.3		13.4 ± 16.9		8.7 ± 7.0		0.402
Medical Treatment ^a^	Topical natamycin	19	(44.2%)	15	(45.5%)	4	(40.0%)	0.761
	Topical voriconazole	19	(44.2%)	15	(45.5%)	4	(40.0%)	0.761
	Topical amphotericin B	17	(39.5%)	14	(42.4%)	3	(30.0%)	0.481
	Oral voriconazole	5	(11.6%)	5	(15.2%)	0	(0.0%)	0.075
	Oral Fluconazole	2	(4.7%)	2	(6.1%)	0	(0.0%)	0.425
	Antifungal usage(average days)	31.8 ± 33.4		35.0 ± 36.7		21.8 ± 17.3		0.282
Surgical treatment	Keratectomy	12	(27.9%)	9	(27.3%)	3	(30.0%)	0.866
	AMT	4	(9.3%)	3	(9.1%)	1	(10.0%)	0.931
	TPK	7	(16.3%)	5	(15.2%)	2	(20.0%)	0.716
Complications	Endophthalmitis	2	(4.7%)	1	(3.0%)	1	(10.0%)	0.359
	Perforation	10	(23.3%)	7	(21.2%)	3	(30.0%)	0.564
Visual outcomes	Median initial logMAR VA (IQR)	1.30	(0.55~2.28)	1.70	(0.70~2.14)	0.47	(0.17~2.20)	0.171
	Median final logMAR VA (IQR)	0.61	(0.28~1.75)	0.70	(0.40~1.0)	0.40	(0.22~1.61)	0.878
	Final VA < 20/200	15	(34.9%)	12	(36.4%)	3	(30.0%)	0.572

^a^ Total percentage of treatment method exceeded 100% because some patients received multiple antifungal treatment or were combined with surgical treatment. AMT: amniotic membrane transplantation; IQR: interquartile range; TPK: therapeutic penetrating keratoplasty; VA: visual acuity.

## Data Availability

Data is contained within the article.

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
