# Peer review of "Fusarium Keratitis in Taiwan: Molecular Identification, Antifungal Susceptibilities, and Clinical Features"

_jof, 2022, doi:10.3390/jof8050476_

Round 1

Reviewer 1 Report

The presented manuscript, entiteled "Fusarium Keratitis in Taiwan: Molecular identification, Anti-fungal Susceptibilities, and Clinical Features" concentrates on the Fusarium spp. isolation, molecular identification, as well as antifungal susceptiblity evaluation of isolates of clinical origin in Taiwan. The manuscript brings a lot of scientific novelty and was designed and performed in a comprehensive way. The authors should check for any spelling mistakes etc. Examples are below:

line 171: "pathgens"

line 204: "species" should not be italiced" 

Author Response

Thank you very much for your comments. We have gone through the whole manuscript to check the spelling, and corrected the typos and the font of species/sp.

Reviewer 2 Report

The authors present a study that identified fungi by molecular identification and antifungal susceptibility. The study also compared Fusarium solani species complex with non-Fusarium solani sp. The methods are described and figures and tables used to describe the results. The limitation of the study are well described.

Author Response

Thank you very much for your comments.

Reviewer 3 Report

This is a series of 43 culture-proven cases of Fusarium keratitis, in Taiwan. The authors performed molecular identification and antifungal susceptibilities of the isolates and investigate the effect of species distribution and susceptibilities on the clinical findings and outcomes. Although the findings are not novel, the paper is well written and adds to the knowledge of the epidemiology and treatment options of this potentially devastating eye infection. The methodology is detailed and well-documented. The most interesting part of the data is the comparison of antifungal susceptibilities between FSSC and non-FSSC. Finally, some interesting data on susceptibilities of the Fusarium spp. to the new azoles eficonazole, lanoconazole and luliconazole are novel, albeit in limited number of cases, and add value to the paper.  

Author Response

Thank you very much for your comments.